# A sensorimotor model shows why a spectral jamming avoidance response does not help bats deal with jamming

Omer Mazar[1]*, Yossi Yovel[1,2]*

[1]Sagol School of Neuroscience, Tel Aviv University, Tel Aviv, Israel; [2]Department of Zoology, Tel Aviv University, Tel Aviv, Israel

**Abstract** For decades, researchers have speculated how echolocating bats deal with masking by conspecific calls when flying in aggregations. To date, only a few attempts have been made to mathematically quantify the probability of jamming, or its effects. We developed a comprehensive sensorimotor predator-prey simulation, modeling numerous bats foraging in proximity. We used this model to examine the effectiveness of a spectral Jamming Avoidance Response (JAR) as a solution for the masking problem. We found that foraging performance deteriorates when bats forage near conspecifics, however, applying a JAR does not improve insect sensing or capture. Because bats constantly adjust their echolocation to the performed task (even when flying alone), further shifting the signals' frequencies does not mitigate jamming. Our simulations explain how bats can hunt successfully in a group despite competition and despite potential masking. This research demonstrates the advantages of a modeling approach when examining a complex biological system.

*For correspondence:
omer_mazar@yahoo.com (OM);
yossiyovel@gmail.com (YY)

**Competing interests:** The authors declare that no competing interests exist.

## Introduction

Echolocation, a prime example of active sensing, provides bats with the ability to detect and hunt flying insects while avoiding obstacles in total darkness (*Griffin, 1953*). Echolocating bats emit high-frequency sound-signals and process the reflected echoes to sense their surroundings. While hunting in a group, conspecific bats emitting calls with similar frequencies may prevent nearby bats from detecting and processing their own echoes. Understanding how bats avoid this process, which is referred to as 'jamming' or 'masking', and how bats segregate the desired weak echoes from the much louder calls emitted by other bats is one of the central debates in the field. We define a 'masking signal' as any signal that reduces the bat's ability to detect and localize an echo due to an overlap with the echo in both time and frequency (*Clark et al., 2009*), and a 'jamming signal' as a signal that completely blocks the detection of an echo. That is, a jamming signal is a masking signal that is more intense than the desired echo (see Materials and methods).

The question of how bats deal with conspecific masking (*Beleyur and Goerlitz, 2019*; *Jones and Conner, 2019*) and whether they perform a spectral Jamming Avoidance Response (JAR) has been widely studied but is still under dispute. Many studies have suggested that bats change their echolocation frequencies when hunting at the presence of other bats (*Takahashi et al., 2014*; *Ibáñez et al., 2004*; *Chiu et al., 2009*; *Ulanovsky et al., 2004*) or when exposed to playback of partially or fully overlapping signals (*Gillam et al., 2007*; *Gillam and Montero, 2016*; *Bates et al., 2008*; *Luo and Moss, 2017*; *Corcoran and Conner, 2014*). In contrast, several recent field-studies and laboratory experiments found no evidence for a use of spectral JAR by bats (*Cvikel et al., 2015a*; *Amichai et al., 2015*; *Götze et al., 2016*). Particularly, in this study, we only deal with *spectral* JAR (which we will term JAR).

The main goal of our study is to use a mathematical approach in order to deepen the understanding of the masking problem and its impact on bats' hunting, and specifically to examine whether an intentional shift of signal frequencies (i.e., a spectral JAR) can assist bats to mitigate the masking problem. We developed an integrated sensorimotor model of bats pursuing prey. The modeling approach entails several advantages in comparison to studies with real bats. (1) It allows us to assess the acoustic input received by each of the hunting bats at every instance. This is currently impossible to do in reality even when a microphone is placed on the bat. (2) Modeling enables manipulation of different parameters and examining their influence on masking including testing hypothetical scenarios that tease apart factors that are coupled in reality.

We analyzed the effect of masking under various prey and bat densities and when using different echolocation behaviors. We measured the probability of jamming, the hunting performance and we explicitly examined whether applying a spectral JAR improves hunting performance when hunting with conspecifics. We were able to discriminate between the effect of direct interference resulting from the need to avoid conspecifics and to compete with them over prey, and the effect of sensory masking due to conspecific calling. We show that shifting the emission frequencies (i.e., a JAR) does not assist mitigating masking because bats' calls already differ from each other due to their well-known behavior of adjusting the echolocation parameters of the calls based on the task and the environment.

## Results

The model consists of numerous bats searching for and attacking prey in a confined 2D area using echolocation. Each simulated bat transmits sound calls and receives the echoes returning from prey items and obstacles, as well as the calls emitted by conspecifics, which might mask or jam its own echoes. The prey's movement mimics a moth (*Stevenson et al., 1995*; *Willis and Arbas, 1991*) with no ability to hear the bats. Prey echoes are detected and localized based on biological-relevant assumptions which consider sound reflection and propagation, and hearing physiology (see Materials and methods). Based on the acoustic input, the bat decides whether to continue searching, to pursue prey or to avoid obstacles such as other bats. The acoustic input is composed of the following signals: (1) insect and obstacle echoes generated by the bat's own calls (i.e. 'desired echoes'), (2) echolocation calls of conspecifics, and (3) echoes returning from calls emitted by conspecifics. The bat then adjusts its echolocation and movement according to the vast literature on bat echolocation (*Griffin, 1953*; *Griffin et al., 1965*), and the recently published control models of bat flight and hunting (*Kalko, 1995*; *Schnitzler et al., 1987*; *Wilson and Moss, 2004*; *Surlykke and Moss, 2000*; *Schnitzler et al., 1988*; *Vanderelst and Peremans, 2018*; *Giuggioli et al., 2015*). For example, the simulated bats emit search calls with a source-level of 110 dB-SPL (at 0.1 m, reference 20 µPa) and they lower it (and adjust other echolocation parameters) when approaching prey. A successful hunt (i.e. a capture) occurs only when the simulated bat gets within 5 cm from the prey. That is, the bats occasionally initiate attacks but miss.

We first demonstrate that our simulations behave similarly to bats. The simulated bats managed to detect, pursue and capture prey at high rates both when hunting alone and when hunting in a group (see *Figure 1*, *Video 1* and *Video 2* for examples of hunting by simulated bats). The movement parameters of the bats in both single and multiple individual scenarios were similar to those of actual bats, flying in a flight room ($4.5 \times 5.5 \times 2.5$ m$^3$), suggesting that our model managed to capture the essence of the foraging movement (*Figure 1—figure supplement 1*). Moreover, the hunting rate of the simulated bats was also comparable to those of real bats. A simulated bat flying in a density of 10 prey items per 100 m$^2$ attacked 5–7 times per 10 s, similar to the rates reported for *P. abramus* (*Fujioka et al., 2014*), a bat with similar foraging behavior, hunting in a comparable environment.

### The influence of a spectral JAR

We next compared the detection and localization performance of bats applying a JAR to bats that do not actively react to masking signals. In both groups, the individuals' terminal frequencies of the FM sweep-calls were sampled from a normal distribution with a standard deviation of 4 kHz, as observed in nature (*Schnitzler et al., 1987*; *Schnitzler et al., 1988*; *Bartonička et al., 2007*), and the bats adjusted their calls according to their task and their distance to object. The JAR was

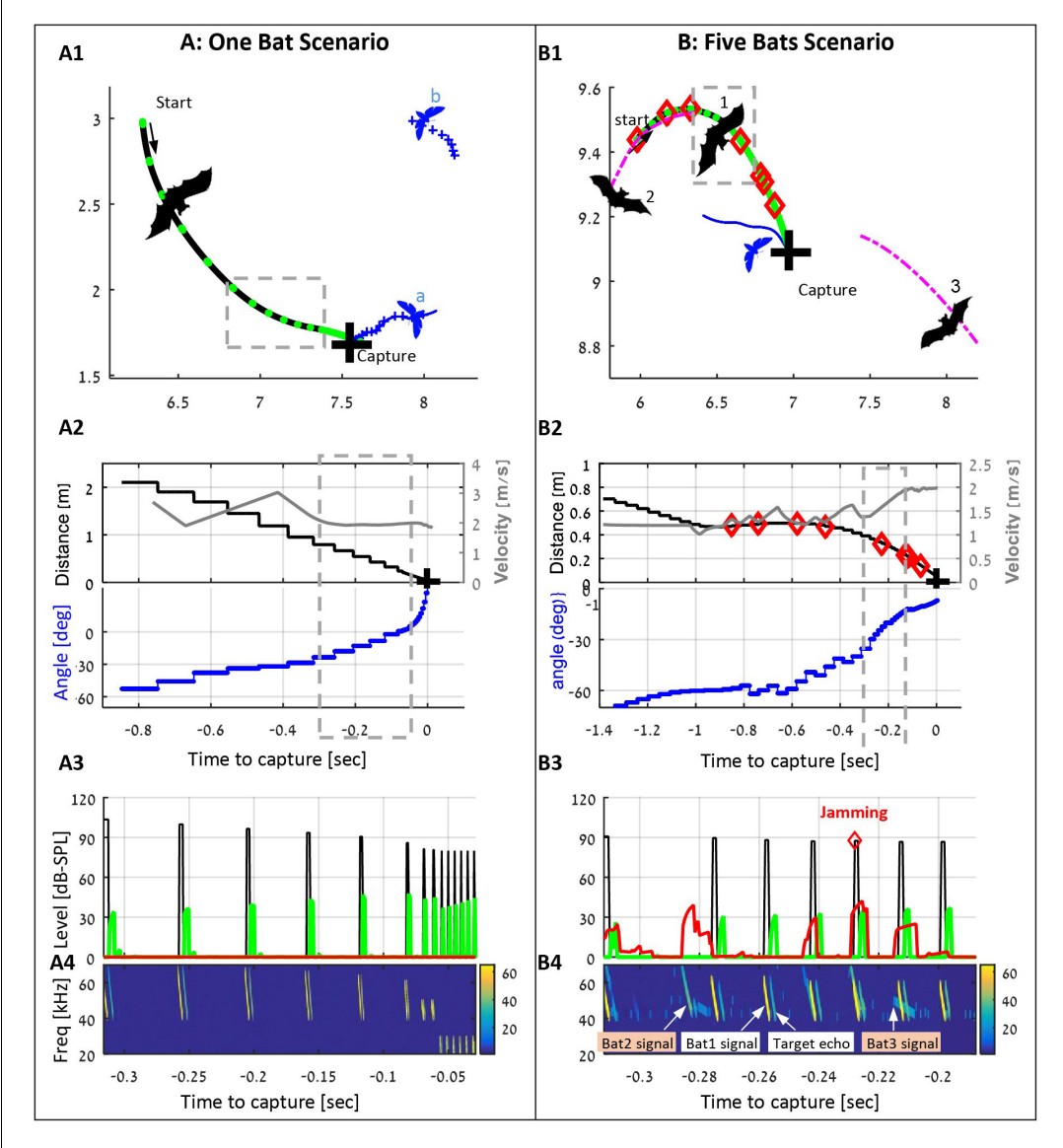

**Figure 1.** examples of individual and group hunting of the simulated bats (also see the *Videos 1* and *2*). (A) A single simulated bat flying in the arena with five prey items. The bat detects two prey items and decides to pursue the closest one (moth a). Panel A1 shows the trajectories of the bat (black) and the moths (blue), the positions of each emitted call (green dots), and the location of the capture (black cross). Panel A2 shows the bat's velocity (gray), the distance to the prey (black) and relative angle (blue) to it. Panel A3 shows the level of the following signals (while the bat is in the gray-bounded area in panel A1): the transmitted calls (black), the received echoes reflecting off the prey items (green), and the received masking signals (red). Panel A4 shows a spectrogram of the same segment as in panel 3. Note how the frequency drops at the final terminal buzz. (B) Three bats (out of five in the arena) hunting in an environment with 10 prey items. Bat one detects and pursues a moth, while conspecifics (bats 2 and 3) are flying nearby and emitting echolocation calls. Some of the conspecifics' calls mask (or jam) the echoes received by bat 1. Instances of jamming are marked by red diamonds in panels B1-B4. All colors and symbols in B1-B4 are the same as in A1-A4. Magenta lines depict trajectories of conspecifics. Panel B4 also demonstrates the variations between the calls of each bat due to the different behavioral phases. Note that bat two detects and pursues the same prey item and thus emits 'approach' calls. From the detection to the successful capture, bat 1 emitted 64 echolocation calls, 8 of which were jammed.

The online version of this article includes the following figure supplement(s) for figure 1:

**Figure supplement 1.** Simulated and real bats flight characteristics.
**Figure supplement 2.** The Bat Module streamlines.
**Figure supplement 3.** Angles and distances for two bats and two prey items.
**Figure supplement 4.** The modified piston model for the directivity of ear and mouth of the bat.

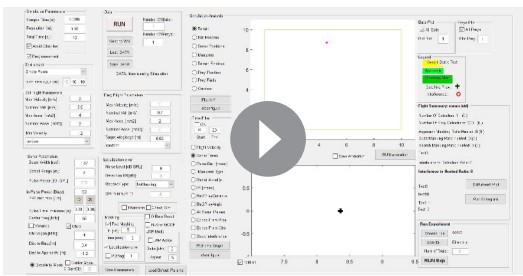

**Video 1.** One bat and one prey item. The video presents one simulated bat (thick blue line) flying in the arena with one prey item (thin dot). The bat starts flying randomly, and whenever it detects the prey it heads towards it and tries to capture it. After each successful capture, indicated by a cross in the video, a new prey item reappear in a random place in the arena.
https://elifesciences.org/articles/55539#video1

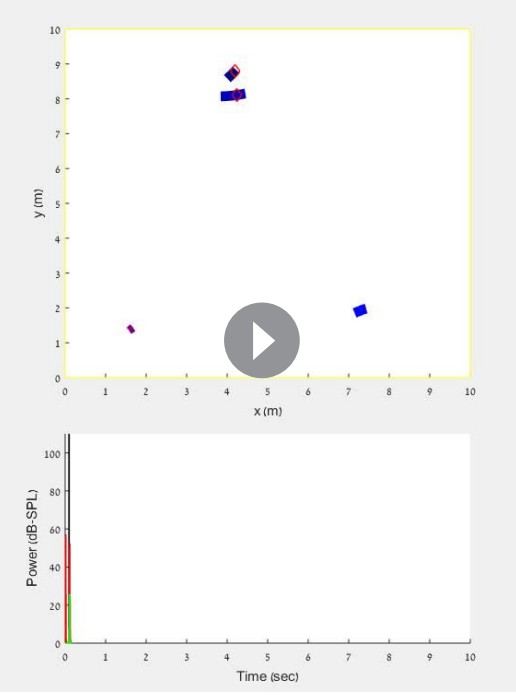

**Video 2.** Three bats with 10 prey items. The upper panel displays three simulated bats (thick lines) flying simultaneously with 10 prey items (colored dots). Thin lines on top of the bats' trajectories indicate the behavioral phase: 'green' for approach, 'pink' for buzz, and 'cyan' for an obstacle-avoidance maneuver (avoiding the arena-borders or other bats). Red diamonds indicate locations and timings of the jamming events (i.e. whenever the desired echo is completely blocked by a masker). The lower panel shows the level (dB-SPL) of the following signals as a function of time: the transmitted calls (black), the echoes reflecting off the prey items (green), and the received masking signals (red). The illustrated signals are acquired from the trajectory of the bat that is depicted by the dark blue line, starting at the upper-left position.
https://elifesciences.org/articles/55539#video2

modeled as follows: whenever the echoes reflected off the closest prey were jammed, the bat shifted its entire frequency range in steps of 2 kHz, upward or downward, to minimize the frequencies' overlap between the masking signal and its own call. If the terminal frequency of the masking signal was higher than its own, the bat shifted the entire frequency-band downward, and vice versa. The bat continued shifting the frequency as long as jamming recurred (Materials and methods).

We tested two different receiver models with different assumptions: 1) the 'correlation-detector' which is an optimal receiver across all frequencies and is at least slightly better than the bat's brain (*Denny, 2004*; *Griffin et al., 1963*; *Erwin et al., 2001*; *Simmons et al., 2004*; *Wiegrebe, 2008*; *Peremans and Hallam, 1998*). (2) The 'filter-bank receiver' which is considered to represent the mammalian auditory physiology and implements a series of gammatone band-pass filters (*Wiegrebe, 2008*; *Boonman and Ostwald, 2007*; *Sanderson et al., 2003*; *Suga, 1990*; *Weissenbacher and Wiegrebe, 2003*)(see Materials and methods).

The JAR has been hypothesized to improve the reception of the desired echo by reducing the overlap between the spectra of the masking signal and the echo. The reception process consists of three main tasks: detection, localization and discrimination of the desired echo. We, therefore, examined three reception-criteria reflecting these three tasks (see Materials and methods): (1) The jamming- probability defined as the probability that the echo reflected off the closest prey item is jammed by a masking signal and is thus not detected. (2) The ranging error defined as the difference between the estimated and the actual distance to prey. (3) The false-alarm rate defined as the probability of identifying a masking signal as prey by mistake. We examined whether a JAR improves reception by comparing these three criteria under different conditions (e.g., different receiver models and different bat densities).

In all scenarios, and for the two different receiver models, the jamming avoidance response did not decrease spectral masking, and did not improve detection performance according to any of the three criteria defined above. See *Figure 2*: One-way ANOVA statistics for correlation and filter-

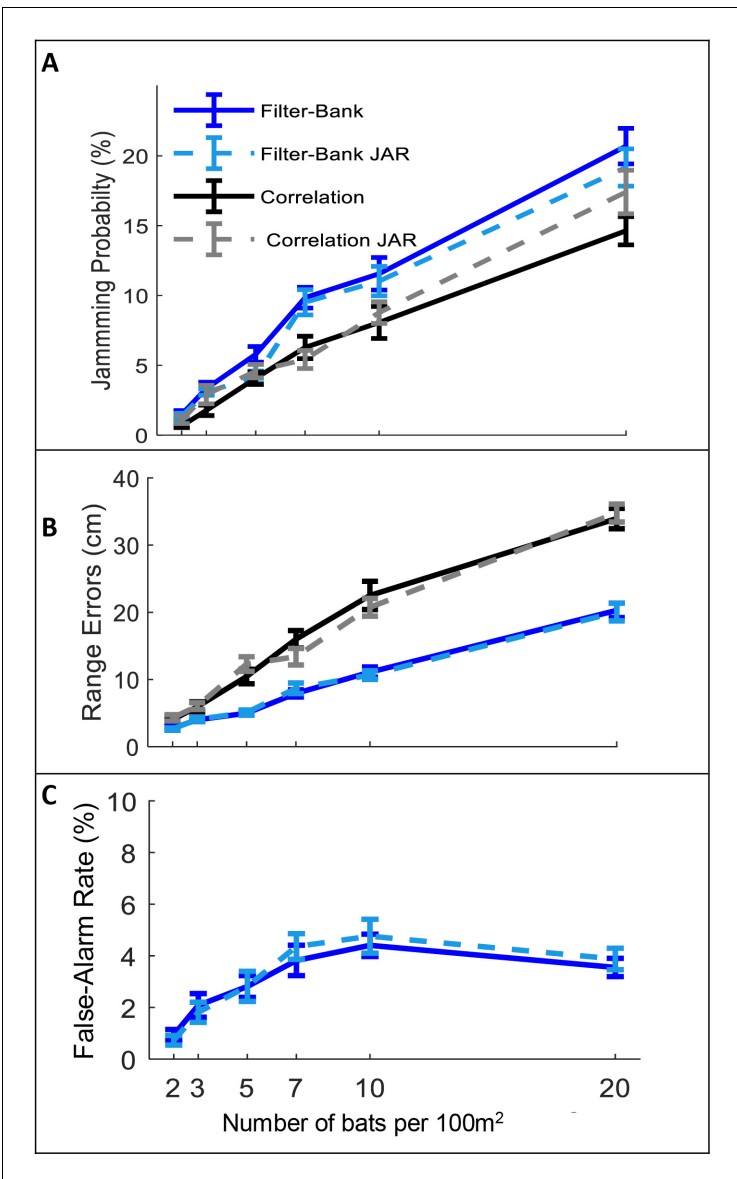

**Figure 2.** The effect of JAR on the receiver. Panels A-C depict three reception criteria as a function of bat density for a correlation-receiver (solid black), and a filter-bank receiver (solid blue) with and without a JAR (dashed gray and dashed turquoise, respectively). The prey density was constant with 20 prey items per 100 m$^2$. (**A**) Jamming Probability. (**B**) Range errors. Note that the range-errors are high because they are calculated for all detected prey. The errors for the pursued prey (i.e. the prey items that bats attacked) are substantially lower, since, as the bats approach their targets echoes have higher SNR and the errors decrease. For the correlation receiver, the range errors are calculated as a function of the SNR (see **equation 9**). For the filter bank receiver, the range errors are derived from the time difference between the estimated time of the detected peak and the arrival time of the desired echo. (**C**) False-alarm rate, which is measured only for the filter-bank model (Materials and methods). Error-bars depict standard-errors in all panels. Each data point represents 60–80 individual bats in each scenario. The online version of this article includes the following source data, source code and figure supplement(s) for figure 2:

**Source code 1.** The source data used to produce **Figure 2**.
**Source data 1.** The data for **Figure 2** and **Figure 2—figure supplement 2**, correlation receiver.
**Source data 2.** The data for **Figure 2**; **Figure 2—figure supplement 2**, filter-bank receiver.
**Figure supplement 1.** Analysis of correlation gains between signals of search (**A**), approach (**B–C**) and buzz (**D**) phases.
**Figure supplement 1—source data 1.** The data for **Figure 2—figure supplement 1**.
*Figure 2 continued on next page*

*Figure 2 continued*

**Figure supplement 2.** Number of prey detection events.

**Figure supplement 3.** Forward and Backward Masking Criteria.

**Figure supplement 4.** Filter-Bank Receiver.

**Figure supplement 5.** Localization errors of the Correlation and Filter-Bank receivers.

**Figure supplement 6.** Implementation of a half-wave rectifier a with 10 kHz LPF (iir filter of order 3 [*Sanderson et al., 2003*]) in the Filter-Bank receiver model did not influence our results: (A) Hunting Performance, (B) Jamming Probability, (C) Ranging Errors, (D) False Alarms.

**Figure supplement 6—source data 1.** The data for *Figure 2—figure supplement 6*, correlation receiver.

**Figure supplement 6—source data 2.** The data for *Figure 2—figure supplement 6*, filter-bank receiver.

---

bank, respectively**: jamming-probability**: $F_{1,237}$ = 2.96, p=0.09; $F_{1,237}$=2.26, p=0.13; $F_{1,237}$ = 0.02 p=0.88, **ranging error**: $F_{1,237}$ = 0.1 , p=0.76; $F_{1,241}$ = 0.01, p=0.96, **false-alarm**: $F_{1,237}$=0.19, p=0.66, for filter-bank only.

A possible explanation for this seemingly surprising result is the fact that a bat's calls continuously vary depending on its behavioral phase and its distance to the targets. Therefore, at any instance, the calls of two bats will already differ, even if their call repertoire is identical. Thus, the influence of additional variability between the calls, achieved by spectral JAR, is insignificant, as we also demonstrate (see *Figure 2—figure supplement 1*).

## The effect of masking on hunting

Next, we tested the effect of masking on hunting performance (i.e., the prey capture rate) under different scenarios. We started with a hypothetical scenario ('no-masking'), in which bats forage in a group without any masking, that is they detect and pursue prey as if there is no sensory masking, but they still have to avoid other bats and sometimes lose prey due to competition. This null-hypothesis scenario enabled us to estimate the non-sensory effects of group hunting, which would be very difficult to do in an experiment with real bats. Ultimately, it allowed us to isolate the effect of sensory masking only. Hunting performance was measured in different bat-densities (from 1 to 20 bats per 100 m$^2$) and different prey densities (3, 10, 20 moths per 100 m$^2$), see *Figure 3*.

Even without any sensory masking (i.e., in the 'no-masking' condition), hunting performance significantly degraded as bat density increased due to competition over prey and due to the need to avoid conspecifics (*Figure 3A–C*; see Green lines). The reduction in performance was significant in all prey densities and resulted in a maximum decrease of 36%, 57% and 67% in performance when the bats' density increased (from 1 to 20) at three prey densities: 3, 10 and 20 prey items per 100 m$^2$, respectively. See *Figure 3* A1–C1, One-way ANOVA, $F_{1, 188}$ = 64.3, p<0.0001; $F_{1, 188}$ = 58.9, p<0.0001; $F_{1, 128}$ = 36.1, p<0.0001, respectively.

We next examined the masking-effect which we defined as the reduction in performance resulting from sensory masking only, relative to the no-masking performance (*Equation 1*).

$$Masking\ effect = 100 * \left( 1 - \frac{\text{Performance with masking for a given scenario}}{\text{Performance without masking}} \right) \quad (1)$$

Sensory masking further hindered hunting under all conditions, but there was no significant difference in performance whether the bats used a JAR or not (*Figure 3*, A1-C1, compare blue, black and magenta lines; ANCOVA, $F_{3, 1218}$ = 2.53, p=0.08, $F_{1, 1581}$ = 0.57, p=0.56, and $F_{1, 1085}$ = 0.24, p=0.78, for 3, 10 and 20 prey items per 100 m$^2$, respectively. This was the case also for another theoretical scenario that we tested: the 'no-frequency variation scenario', in which all bats had the same call repertoire and, hence, it simulates an extreme case with maximal sensory masking (One-way ANOVA, between 'no-frequency variation' and JAR with random frequencies: $F_{2,856}$=2.74, p=0.1; $F_{2,971}$=0.85, p=0.36; $F_{2,1086}$=0.43, p=0.64, for prey densities of 3,10 and 20 prey items per 100 m$^2$, respectively). See *Figure 3—figure supplement 1* for a similar analysis using the filter-bank model.

To deepen our understanding of why a JAR does not improve performance, we analyzed the jamming-probability in different behavioral phases. We found that jamming mostly occurred during the search phase while, as the bats shifted from the search to the approach phase, the probability of jamming decreased significantly because the prey's echoes become louder (*Figure 3* A3–B3–C3,

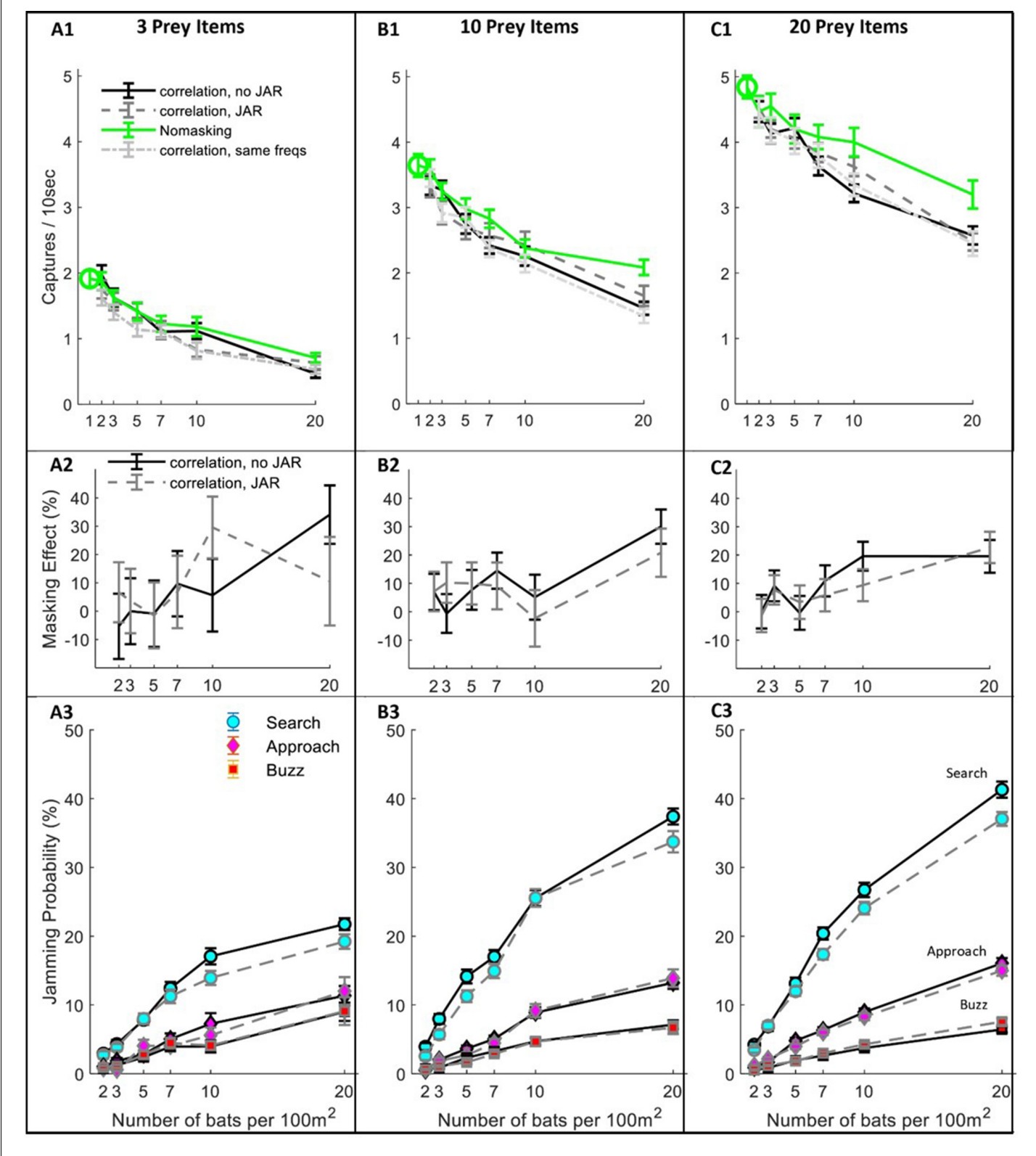

**Figure 3.** Hunting Performance for different receiving models. The hunting success rate (A1–C1), masking effect (A2–C2) and jamming probability (A3–C3) are presented for three prey densities per 100 m$^2$: three prey items, 10 prey items, 20 prey items. Panels A1-C1 depict the performance as a function of bat density; a green circle shows the performance of a single bat. Line colors and styles depict the performance of different receiver models: solid green - no-masking; solid black - correlation-detector with random frequency variability and without JAR; dashed dark gray - correlation-detector

*Figure 3 continued on next page*

*Figure 3 continued*

and random frequency with JAR response; dash-dotted light gray - correlation-detector without frequency variability. The regression slopes of no masking condition (green lines) are (mean ± SD): 0.06 ± 0.0065 0.08 ± 0.0084, 0.079 ± 0.013 captures per bat per ten seconds, at the prey densities above, respectively (ANCOVA: $F_{1, 564}$ = 84.3, p<0.0001; $F_{1, 679}$ = 90.4, p<0.0001; $F_{1, 431}$ = 37.8, p<0.0001). There was no significant difference in performance when applying or not applying spectral JAR - see main text and compare gray and black lines. Panels A2-C2 show the masking-effect on hunting, that is the relative decrease in hunting relative to the 'no-masking' condition. Panels A3-C3 present the probability of jamming during the behavioral phases: search (turquoise marker), approach (magenta marker) and buzz (red markers). Jamming probabilities during the search phase were significantly lower by at most 4.5% when using a JAR (ANCOVA, $F_{1, 2422}$ = 23.42, p<0.0001). However, in the approach and buzz phases (which are more critical for foraging), there was no significant difference between the two models (ANCOVA, $F_{1,2388}$ = 0.11, p=0.74; $F_{1, 2347}$ = 0.11, p=0.73, respectively). Error bars show means and standard-errors for 70–120 simulated bats in each data-point.

The online version of this article includes the following source data, source code and figure supplement(s) for figure 3:

**Source code 1.** the source data used to produce *Figure 3*, panels A1-C1.
**Source code 2.** the source data used to produce *Figure 3*, panels A2-C2.
**Source code 3.** the source data used to produce *Figure 3*, panels A2-C2.
**Source data 1.** The data for *Figure 3*, panels A1-A3.
**Source data 2.** The data for *Figure 3*, panels B1-B3.
**Source data 3.** The data for *Figure 3*, panels C1-C3.
**Figure supplement 1.** Hunting performance for the filter-bank receiver.
**Figure supplement 1—source data 1.** The data for *Figure 3—figure supplement 1*, panels A1-A3.
**Figure supplement 1—source data 2.** The data for *Figure 3—figure supplement 1*, panels B1-B3.
**Figure supplement 1—source data 3.** The data for *Figure 3—figure supplement 1*, panels C1-C3.
**Figure supplement 2.** The influence of background clutter echoes.
**Figure supplement 2—source data 1.** The influence of background clutter echoes.
**Figure supplement 3.** The causes of hunting failures.

ANCOVA, for the comparison between search and approach at any bat and prey density: $F_{1, 8527}$ = 2784, p<0.0001, panels B3-C3 show that less than 15% of the prey echoes are jammed during the approach phase). Notably, jamming during the search phase is less influential than it might seem to be because, if a potential prey echo is jammed, the bat is likely to detect the prey with one of its following emissions. The low probability of jamming during the approach is probably the main reason for the relatively small effect of sensory masking on performance.

We further explored whether the presence of background clutter echoes would modify our results, giving an advantage to a JAR. In a set of clutter simulations, we took into account the echoes reflected off the border of the arena as if the bats were foraging in a forest opening surrounded by vegetation (see Materials and methods). We modeled the received levels and timings of the clutter echoes reflected from the background assuming several target strengths. We conclude that also in a cluttered environment applying a JAR had no significant effect on the hunting performance, *Figure 3—figure supplement 2*. As expected, hunting performance decreased as the level of the clutter echoes increased.

We also analyzed the causes of unsuccessful attacks when bats initiated an attack but failed to capture prey. There were four reasons for failed attacks: avoiding collisions with a nearby conspecifics, losing the prey to a conspecific, avoiding an obstacle (the borders of the arena) and missing the prey due to an insufficient maneuver or due to sensory error (resulting for example, from jamming, hereafter 'Misses'). We analyzed the proportion of these different sources of failure with and without sensory masking (see *Figure 3—figure supplement 3*). With 20 bats and 10 prey items per 100 m$^2$, without masking, 34 ± 2% of the capture attempts were successful (mean ± SE). The unsuccessful attempts were due to conspecifics avoidance: 27 ± 2%; lost prey to conspecifics: 17 ± 1.5%; obstacle avoidance: 7 ± 2% and misses: 15 ± 2%. When sensory masking was added, the proportion of successful captures significantly decreased to 26 ± 2% (One-way ANOVA, $F_{1, 198}$=4.59, p=0.033), and misses became the most substantial cause for failure, significantly increasing to 38 ± 3.5% (One-way ANOVA: $F_{1, 198}$=68.8, p<0.0001). The total number of hunting attempts, however, was not affected by the masking (effect-size = 0.01 trials per 10 s; one-way ANOVA: $F_{1, 198}$=0.08, p=0.97).

## The effect of the echolocation signal design and the detection threshold on hunting in a group

After observing that spectral changes do not assist mitigating jamming, we tested whether other adjustments to echolocation or physiological parameters could improve bats' performance when hunting in a group. We tested the effect of three prime parameters: source level, call duration, and detection threshold (or hearing sensitivity), which is a function of the auditory system and could, in theory, be changed by evolution. For all parameters, we used a range of values suggested in the bat literature. For each of these parameters, we first examined its effect on the overall hunting success when foraging in a group (i.e., including both direct competition and sensory masking), and we then examined the parameter's effect specifically on the masking.

Increasing the source level improved hunting performance, but only up to a level of ca. 110–120 dB-SPL, (at 0.1 m) above which the improvement was negligible and insignificant (*Figure 4* A1, shows that performance increased significantly when source level increased from 90 to 110 dB: ANCOVA, slope = 0.06 captures/dB, $F_{4,\ 3815}$=386, p<0.001; performance did not change when source level increased from 120 to 150 dB: $F_{4,\ 4795}$=0.1, p=0.98). Interestingly, the source level did not affect the masking effect (i.e., the reduction in performance beyond the no-masking condition), suggesting that increasing the emission level assists hunting in general (through increasing the detection range) and does not assist overcoming the masking problem specifically (*Figure 4* A2; F = 2.45, p=0.16, Pearson linear regression).

Changing the duration of search-calls had no significant effect on the hunting performance (*Figure 4* B1: ANCOVA, $F_{1,\ 2314}$=0.15, p=0.69). To test the effect of call duration we slightly varied the simulation (see the detection section in the Materials and methods).

Decreasing the hearing threshold (under a constant ambient noise level) significantly improved hunting (*Figure 4* C1, ANCOVA, $F_{4,\ 2395}$=915, p<0.0001). Like in the case of increasing emission level, changing the hearing threshold did not significantly change the masking effect (*Figure 4* C2; ANCOVA, $F_{1,\ 797}$=2.19, p=0.14). This is probably because the decrease in the hearing threshold increases the probability of detecting both echoes and masking signals. With a high hearing threshold of 30 dB and a density of 5 bats per 100 m$^2$, there seems to be a negative masking effect, but that is because prey is only detected from very short distances and thus prey detection and masking hardly occur, and consequently, the standard error of our estimate under these conditions is high.

## Discussion

The jamming problem is one of the most fundamental challenges raised by researchers of echolocation, but, only a few studies (*Beleyur and Goerlitz, 2019*; *Lin and Abaid, 2015*; *Jarvis et al., 2013*; *Cvikel et al., 2015b*) used a mathematical model to examine the actual chances of being jammed by another bat, and how such jamming would affect hunting performance. Addressing these questions is a difficult task with real bats as even if a microphone is placed on the bat, it is typically not as sensitive as the bat itself and it is not placed inside the ear. The substantial body of literature that has accumulated on bat echolocation and sensorimotor control now allows simulating natural scenarios where bats are foraging in aggregations. Using this approach, our simulations suggest how even in very high bat-densities, bats can probably capture insects at high rates. Because our model follows a conservative approach underestimating the bats' performance (see Materials and methods), this result likely reflects the maximum impact of jamming. Indeed, bats' ability to hunt and avoid obstacles in high conspecific density has been documented (*Cvikel et al., 2015b*). Notably, we did not fit any of the model's parameters – we used parameters that are based on our measurements on real bats or published results. Similarly, we used a simple control strategy to steer the bat to the prey. For example, we do not assume any memory of the position of the target, nevertheless, our simulated bats manage to catch prey even if some information is degraded or completely missing due to jamming. Real bats probably use memory to overcome temporal miss-detections caused by momentary jamming and are thus probably better than our simulated bats. Furthermore, our analysis is based on relative measurements between different scenarios, therefore, even if the exact rates of prey-capture that we estimated are biased, the principles which we observed are likely correct, providing insight to the jamming problem.

The two most important results are: (1) Much of the interference that bats suffer from when foraging in a group results from competition over prey and from the need to avoid conspecifics, and not

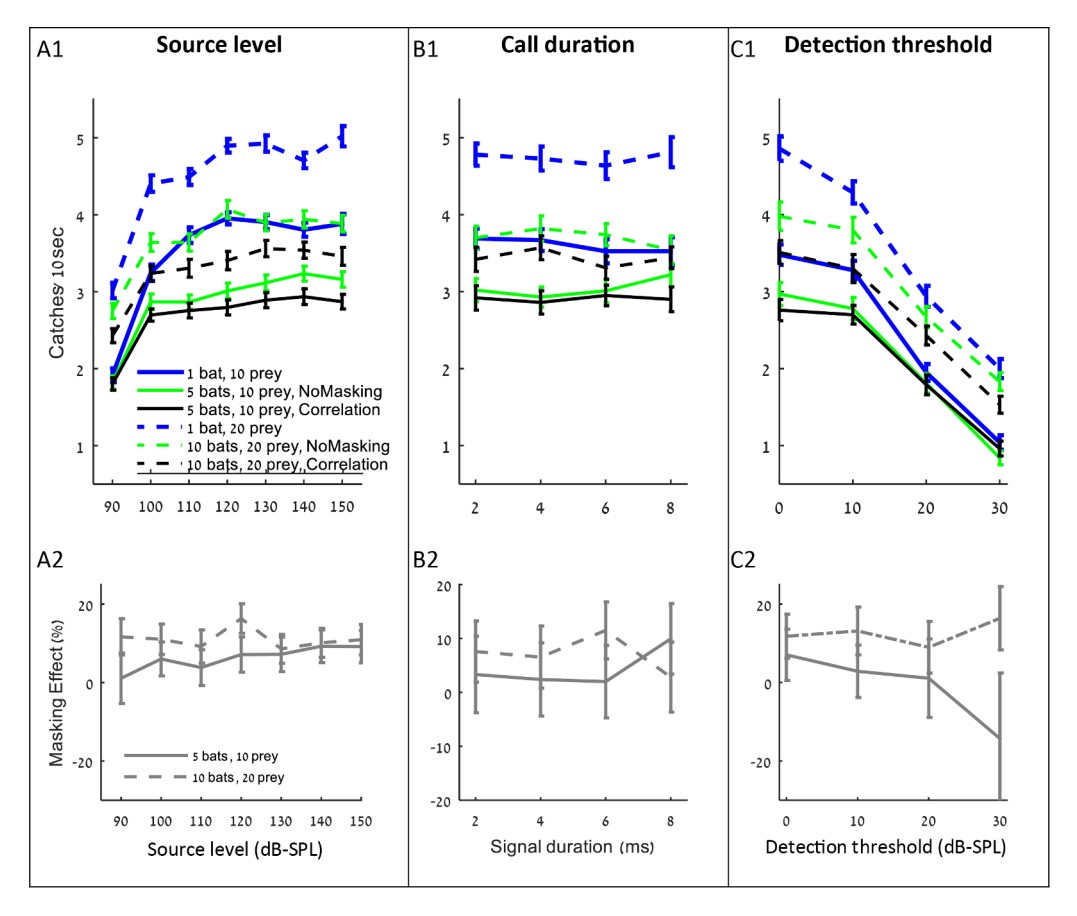

**Figure 4.** The influence of source level, call duration and detection threshold on the performance (**A1–C1**), and masking effect (**A2–C2**). The panels include the following conditions: the correlation-detector with frequencies normally distributed (black), no-masking (green), and one bat (blue). Solid lines indicate a density of 10 prey items while dashed lines represent scenarios with 20 prey.

The online version of this article includes the following source data, source code and figure supplement(s) for figure 4:

**Source code 1.** The source data used to produce *Figure 3*, panel A1.
**Source code 2.** The source data used to produce *Figure 3*, panel B1.
**Source code 3.** The source data used to produce *Figure 3*, panel C1.
**Source data 1.** The data for *Figure 4* panels A1-A2.
**Source data 2.** The data for *Figure 4* panels B1-B2.
**Source data 3.** The data for *Figure 4* panels C1-C2.
**Figure supplement 1.** Frequency- shift with no JAR.
**Figure supplement 1—source data 1.** The data for *Figure 4—figure supplement 1*.

from acoustic masking. One of the main reasons for this is that jamming mostly occurs when the bat is still searching for prey, while once it has detected prey and is closing in on it, prey echoes become loud, and the chances of jamming substantially decrease. Masking during search might sound problematic, but even if a prey's echo is completely jammed, another echo from the same prey will likely be detected with one of the following echolocation calls. (2) Using a spectral JAR, which has been suggested by many previous researchers, is ineffective for solving the jamming problem or even reducing it. The reason for this is that bats constantly change their signals according to behavioral phase and distance to nearby objects. Even if two bats have the same call repertoire, at any moment in time, their calls are different due to their different behavioral phase and because they are likely to have objects at different distances. Moreover, we only used the bats' first harmonic. Simulating the second harmonic too, thus using signals with more than twice as much bandwidth, would have probably made the jamming avoidance response even more irrelevant, because the differences between

the calls of different bats would be naturally larger at any moment (even without a JAR). In theory, in bats that emit narrow bandwidth calls, such as bats with shallow FM calls (*Schnitzler et al., 2003*), jamming might be more influential and a spectral JAR might be more beneficial. However, most shallow FM bats increase bandwidth considerably when pursuing prey, and thus spectral JAR is probably not substantial for those bats too, at least during the pursuit. Indeed, in a previous study, we did not observe a spectral JAR in a bat that uses shallow FM calls (*R. microphyllum*) (*Cvikel et al., 2015a*).

Both of the receiver models that we tested revealed the same results regarding the inefficiency of a JAR. One of the models (the correlation-receiver) is considered optimal in terms of its detection abilities and probably over-estimates bats' abilities. The fact that such a detector, which is extremely sensitive to the specific spectro-temporal pattern of the desired signal, did not show better performance when a JAR was applied, strongly suggests that the JAR is not helpful for real bats too, as their ability to use the differences induced by a JAR are lesser, compared to this receiver.

As expected, the correlation-receiver outperformed the filter-bank in all scenarios: it detects more objects (see *Figure 2—figure supplement 2*) and it has a lower probability to be jammed by masking signals (see *Figure 2A,B*). Consequently, the total hunting performance is better (compare *Figure 3* A1–A3 with *Figure 3—figure supplements 1* and *3* A1-A3). The range-errors of the filter-bank seem lower (*Figure 2c*), but this is only because the correlation-receiver detects farther objects than the filter-bank receiver and these objects have lower SNR and thus higher range errors. This larger error has a little effect on the performance because when the bats get closer to targets, the SNR improves and the errors decrease.

Another interesting result of the simulations was revealed when testing which of the echolocation parameters would allow bats to perform best when hunting in aggregations. We found that the source level actually used by real bats when hunting in a group (ca. 110–120 dB-SPL [*Kalko, 1995*; *Boonman et al., 2013*; *Kober and Schnitzler, 1990*]) gave the best performance in the simulation. Increasing the source level mainly helped increasing prey detection range and not overcoming masking - the masking effect was the same independently of the source level. Moreover, increasing the source level beyond 120 dB-SPL did not further improve hunting, probably because when hunting in aggregations there is no benefit in detecting prey beyond a certain distance. Prey that is farther than this distance is very likely to be detected and caught by a closer bat. In a previous study, we found that when hunting under masking background bats increase call intensity (*Amichai et al., 2015*) and others have described similar results (*Takahashi et al., 2014*; *Luo et al., 2015*). This result demonstrates how all bats can call louder up to a certain degree (i.e. 120 dB-SPL) and still benefit from better performance. It would have been difficult to explain the benefit of everyone calling louder without a simulation (note that we did not consider the caloric cost of increasing the source level which might further reduce the actual level emitted by real bats).

Changing the call's duration did not affect the performance. A possible explanation for this is the fact that all simulated bats increased their call duration. Therefore, the benefit of own longer calls is apparently balanced with the greater probability of overlapping with conspecific signals. Note that we are not saying that call duration is irrelevant for hunting in general, but only that it does not affect the ability to mitigate masking and does not improve hunting in a group.

Why then do bats exhibit JAR-like behaviors? Several previous studies reported that bats change their emission frequencies in response to nearby conspecifics (*Takahashi et al., 2014*; *Ibáñez et al., 2004*; *Chiu et al., 2009*; *Ulanovsky et al., 2004*) or to the playback of masking signals (*Jones and Conner, 2019*; *Gillam et al., 2007*; *Gillam and Montero, 2016*; *Bates et al., 2008*; *Luo and Moss, 2017*). Researchers have interpreted this behavior as a spectral jamming avoidance response. Explaining all previous studies would require much more than a short discussion. We will thus suggest two alternative hypotheses that could explain these findings and should be further pursued. Except for a few exceptions (*Bates et al., 2008*; *Luo and Moss, 2017*), the great majority of previous studies reported an upward frequency shift, that is bats always elevated their frequency. Such a response could be part of the clutter response that is typical for bats when flying in the vicinity of nearby objects. The function of the clutter response (*Hiryu et al., 2010*) is to improve localization of nearby objects; in this case other bats. A clutter response is characterized by emitting calls with higher frequencies and by additional signal adjustments such as a decrease in call duration, as some of these studies indeed reported (*Ulanovsky et al., 2004*). Some of the previous studies were playback experiments (*Gillam et al., 2007*; *Gillam and Montero, 2016*; *Bates et al., 2008*;

*Luo and Moss, 2017*), in which additional bats were not present. In these experiments the clutter should not have increased and thus should not have caused a frequency shift. One possibility is that bats approached the playback speaker (as many bats do [*Cvikel et al., 2015b*; *Dechmann et al., 2009*; *Barclay, 1982*]) and thus clutter actually increased in these experiments. Another possible explanation for the bats' apparent response is based on the Lombard effect, that is the effect of raising source level in the presence of noise, which is well documented in many mammalian species (*Luo et al., 2015*; *Brumm and Zollinger, 2011*; *Zollinger and Brumm, 2011*; *Brumm, 2004*; *Scheifele et al., 2005*), including bats (*Takahashi et al., 2014*; *Amichai et al., 2015*). It is also known that increasing the emission frequencies could be a by-product of the increase in amplitude (*Titze, 1989*; *Genzel et al., 2019*; *Hotchkin and Parks, 2013*). In both hypotheses, the change in frequency does not aim to decrease spectral overlap and thus cannot be considered a spectral jamming avoidance response. Note that other explanations have been suggested for frequency shifts such as solving the ambiguity problem in the presence of clutter (*Hiryu et al., 2010*), and the execution of audio-vocal feedback during the emission period (*Luo and Moss, 2017*).

We used our simulations to test the increased clutter hypothesis by reproducing the analysis performed in previous studies. That is, we analyzed the frequencies used by bats when flying alone and when flying with nearby conspecifics, assuming that the bats did not use a JAR. We then compared the emission frequencies used by solitary bats and by bats in aggregations. Indeed, our simulations show that bats' average frequency in the presence of conspecifics would rise by as much as 400 Hz (as reported in previous studies) in comparison to when flying alone, although they are not performing a jamming avoidance response (see *Figure 4—figure supplement 1*). This result is well-aligned with the findings of *Götze et al., 2016.* that during encounters with conspecifics the terminal frequencies of nearly all calls were within the predicted transmission repertoire of the individual bats. Our results thus provide an alternative explanation for the findings of many of the previous studies that reported a JAR.

Our work demonstrates the power of simulations to reveal new insight into complex biological systems that are difficult to examine and analyze otherwise. Our model shows that jamming is less of a problem than previously suggested by most researchers (but see *Beleyur and Goerlitz, 2019*). It proves that bats can successfully hunt in the presence of other bats without applying any JAR and shows that applying a JAR has no significant impact on hunting performance and on prey detection. Similar (modified) simulations can be used in the future to examine many additional fundamental questions in echolocation and to provide insight that may allow us to interpret previous behavioral results and to design better behavioral studies.

# Materials and methods

## Key resources table

| Reagent type (species) or resource | Designation | Source or reference | Identifiers | Additional information |
|---|---|---|---|---|
| Software, algorithm | MATLAB | MATLAB R2018b. MathWorks | | |
| Software, algorithm | Model; Simulation | This paper | | Newly created using MATALB. See Materials and methods. |
| Software, algorithm | JMP14 | JMP14 Statistical Discovery. From SAS | | |

## General

The MATLAB model simulates the flight and echolocation behavior of *Pipistrellus kuhlii* bats. This small insectivorous bat (approximately 5–9 g) is common in the Mediterranean region and is often observed in groups of ~5 individuals foraging around a street-light (*Amichai et al., 2015*; *Kalko, 1995*; *Schnitzler et al., 1987*; *Barak and Yom-Tov, 1989*). Our hunting-ground is a 10 × 10m$^2$ 2D area with no obstacles. Our model consists of three major modules: the prey module, the bat module, and the acoustics module. The prey module controls the flight maneuvers of the

simulated moths. The bat module simulates the bat's behavior and executes the following processes: decision making, echolocation behavior, flight control, and sensory processing. The acoustics module calculates the received level and timings of all the signals composing the acoustic input (i.e. desired echoes, conspecific calls, and echoes generated by conspecific calls).

## The prey module

The movement of the targets was simulated by a 'correlated random walk' model, resembling a flight path of a moth (*Stevenson et al., 1995*; *Willis and Arbas, 1991*). The linear velocity has an average of 1m/s and it changes every 100ms according to *Equation 2*, where the velocity's change ($v$) is sampled from a normal distribution: 0±0.1m/s (mean± SD, standard deviation). The velocity is bounded between 0.8 and 1.2m/s. The flight direction is determined according to *Equation 3*. In this equation,$\theta_n$, the angular velocity during a 100ms section is a normal random variable with distribution: 0±2 *rad*/s (mean± SD), and τ is the sample time of the model (0.5 ms). The starting position of each moth is drawn from a uniform distribution across the 2D area, its initial flight-direction is random between 0 to $2\pi$ radians, and the starting velocity is also normally distributed: 1±0.1m/s (mean± SD).

$$v_p(n+1) = v_p(n) + v \tag{2}$$

$$\theta_p(t+\tau) = \theta_p(t) + \dot{\theta}_n \cdot \tau \tag{3}$$

The simulated prey does not detect or respond to the pursuing bats. When a moth reaches the borders of the confined area, it changes its flight angle by $\frac{\pi}{2} rad$ relative to the border (to return into the foraging area). To keep a constant prey-density during the simulation, each time a bat captures a moth, a new moth is added to the environment at a random position with a random flight direction.

## The bat module
### Decision making

The echolocation behavior and flight-control of the simulated bats are illustrated in *Figure 1—figure supplement 2*. At the beginning of a simulation, each bat starts foraging in a random position and transmits echolocation calls with 'search' phase parameters (*Table 1*).

After emitting an echolocation call, the bat processes the acoustic inputs (including all echoes and masking sounds, see above) and decides its next step. The rules of the decision making are as follows: (1) If the bat's flight-path comes too close to another bat (less than 20 cm) or the borders of the area, it avoids them and changes its flight direction and velocity. (2) If one or more prey items are detected, the bat chooses the closest one and executes a hunting maneuver. (3) Else, the bat continues searching. According to its decision, the bat adjusts its flight control and echolocation

**Table 1.** the echolocation parameters in the different hunting phases.
Once prey is detected, the hunting phase is defined by the distance to the target. Based on Table 1. During each behavioral phase, the IPI, call duration, bandwidth, and level (in dB) are reduced linearly between the start and end values (Table 1).

| Flight phase | Search | Approach | | Buzz | | |
|---|---|---|---|---|---|---|
| Parameter | | Start | End | Terminal 1 start | Terminal 1 end | Terminal 2 |
| Inter Pulse Interval [ms] | 100 | 70 | 35 | 18 | 6 | 5 |
| Call Duration [ms] | 7 | 5 | 2 | 2 | 1 | 0.5 |
| Terminal Frequency [kHz] | 39 | 39 | 39 | 39 | 39 | 19 |
| Chirp Bandwidth [kHz] | 8 | 35 | 30 | 30 | 20 | 20 |
| Source Level [dB-SPL] | 110 | 110 | 90 | 90 | 80 | 80 |
| Distance to target [m] | >1.2 | 1.2 | 0.4 | 0.4 | 0.2 | <0.2 |

behavior. This decision process is executed every inter-pulse-interval (i.e., between the emissions of two echolocation calls).

## Echolocation behavior

The echolocating behavior of the simulated bats was modeled based on a rich body of literature (*Schnitzler et al., 1987*; *Wilson and Moss, 2004*; *Kober and Schnitzler, 1990*; *Kuc, 1994*). The foraging behavior of insectivorous bats is divided into three main phases: 'Search', 'Approach', and 'Buzz' (*Griffin et al., 1965*). Each phase is characterized by a different set of echolocation parameters. Our model follows the echolocation and hunting behavior of *Pipistrellus kuhlii* based on field studies (*Kalko, 1995*; *Schnitzler et al., 1987*). The bats in the simulation emit Frequency Modulated (FM) down-sweep signals (mimicking *P. kuhliis'* first harmonic, see *Table 1*).

## Alternative *JAR models*

Bats emit linear frequency modulated (FM) down-sweeps. We tested three versions of the model: (1) all bats have the same baseline call, meaning that if two bats are in the same phase and equal distances to targets, their calls will be identical. (2) The bats' terminal frequency was sampled from a normal distribution, with a mean set to 39 kHz, and a standard deviation of 4 kHz in the 'Search' phase. The bandwidth of the calls is constant between bats, so the entire frequency range shifts according to the terminal frequency. This frequency range is in line with the variance of the terminal frequencies reported in the field for this species (*Schnitzler et al., 1987*). (3) To examine the effect of JAR in the third version of the model, bats used active JAR. They evaluated whether their echoes were jammed (i.e., the masking signal blocked the detection of the closest prey). In such cases, they shifted their terminal frequency upward or downward in steps of 2 kHz, to reduce the overlap with the masking signal (i.e. if the masker's frequency was lower than their own, they would raise their frequency). These frequency-shifts are in line with the findings of studies reporting evidence of JAR (*Ulanovsky et al., 2004*; *Gillam et al., 2007*). The bats kept transmitting the modified call for five consecutive calls, and if during that period another echo was jammed, the bats shifted their frequency again in the proper direction. The terminal frequencies were bounded between 35 and 43 kHz (i.e., there was no shift beyond these boundaries). Real bats could, in theory, sense jamming based on several cues. for example during prey pursuit, a bat has expectations regarding the level and the direction of a received echo and a jamming signal would probably violet these expectations. Furthermore, during a search phase, even before the first echo from the target is detected, bats could potentially sense jamming signals based on their spectra which are different from those of reflected echoes, and perhaps also based on the angle of the sound-source which can be outside the echolocation field of view.

## Flight Control

Before a prey is detected, simulated bats fly according to a 'correlated random walk' path, with a constant linear velocity 3.5 m/s and a random change of direction, sampled from a normal distribution of angular velocities: $0 \pm 1$ rad/sec (mean ± SD) (*Vanderelst and Peremans, 2018*; *Erwin et al., 2001*; *Kuc, 1994*). A new angular velocity is sampled before each echolocation emission and the bat turns according to this velocity until the next emission. Once a target is detected the bat turns toward the prey by changing its angular velocity according to its relative direction to the target, using a delayed linear adaptive law described in *Vanderelst and Peremans, 2018*; *Ghose, 2006*.

This dependency is described in *Equation 4*, where $\dot{\theta}_{bat}(t + \tau)$ is the angular velocity of the bat in the next time-sample (i.e., time $t + \tau$), $k_r$ is a gain coefficient, limited by the maximal acceleration of the bat (set to 4 m/s$^2$), and $\emptyset_{target}(t)$ is the current angle between the target and the bat.

$$\dot{\theta}_{bat}(t + \tau) = k_r \cdot \phi_{target}(t) \tag{4}$$

We neglect head movements; the original model (*Vanderelst and Peremans, 2018*; *Ghose, 2006*; *Falk et al., 2014*) refers to the gaze angle (i.e., the angle between the head's direction and the target), but for simplicity, we assume that the head and body are aligned. Even though we know the head and body are not always aligned; this assumption does not affect the behaviors tested in this study. To keep its direction aligned with the target, the bat typically slows down when

the angle to the target ($\phi_{target}$) is large and accelerates linearly when the target is straight ahead (*Ghose et al., 2006*; *Jones and Rayner, 1991*). To model this, we implemented a velocity-model suggested by Vanderelst and Peremans to simulate this behavior (*Vanderelst and Peremans, 2018*), described in *Equation 5*. $V_{phase}$ is the maximal velocity in each behavioral phase (3.5m/s in the approach and search phases, and 2m/s during the buzz phase). Like the direction, the bat adjusts its velocity after each inter-pulse-interval. Indeed, the accelerations and turning rates of the simulated bats correspond well to those reported in the field (*Ghose et al., 2006*; *Jones and Rayner, 1991*), see *Figure 1—figure supplement 1*.

$$V_{bat}(t + \tau) = V_{phase} \cdot cos(\phi_{target}(t)) \tag{5}$$

A successful hunt (capture) is achieved when the bat is less than 5 cm from the prey (*Vanderelst and Peremans, 2018*).

We validated our flight model against the flight trajectories of real bats. We used the movement of three P. kuhli bats recorded in *Taub and Yovel, 2020* trained to search for and land on a static target in a flight room (4.5 × 5.5×2.5 m$^3$). Then, we performed a simulation with three bats in a 5 × 5m$^2$ arena with one static prey item and compared between the flight trajectories and movement parameters. The flight paths, linear velocities, angular velocities, and accelerations of the model were similar to those of the actual bats (*Figure 1—figure supplement 1*) implying that our model represents the foraging movement.

## Sensory processing

The simulated bat detects and estimates the range and direction of objects in the environment (prey and obstacles), based on the incoming acoustic input. Echoes will only be processed if they cross the auditory threshold set to 0 dB-SPL based on the literature (*Boonman et al., 2013*; *Poussin and Simmons, 1982*; *Coles et al., 1989*; *Popper and Fay, 1995*) (we also tested the influence of that threshold between 0–30 dB-SPL, see results, *Figure 4C*). We define such detected echoes as 'Pre-Masking Echoes'. Next, we calculate the effect of masking using two different detection-models: the correlation-receiver which is a well-studied theoretical reference model, and the gammatone filter-bank receiver which represents the temporal reaction of the inner ear to auditory signals. After the preliminary detection, the bat chooses its target again, from the non-jammed echoes.

The **correlation-receiver** is based on a similarity between the bat's own transmitted calls and the received signals (*Denny, 2004*; *Saillant et al., 1993*). The detector calculates two correlations: (i) the self-correlation between the echo and its own transmitted call. (ii) The cross-correlation between the masking signal and its transmitted call. For the echo to be detected, the self-correlation peak should be higher than the cross-correlation peak with more than the 'forward detection threshold' (set to 5 dB) if the cross-correlation peak is within 3 ms before the echo, and higher than the masking peak by more than the 'backward detection threshold' (e.g. 0 db), if the masking signal arrives within 1 ms after the desired echo (see *Figure 2—figure supplement 3*). The periods and thresholds were defined according to 'the law of first wave-front' (*Popper and Fay, 1995*) ch. 2.4.5, (*Blauert, 1997*) ch. 3.1 and (*Mhl and Surlykke, 1989*), and comprise lower boundaries of real bats' abilities to cope with masking (*Beleyur and Goerlitz, 2019*). For simplicity, we used constant thresholds within each window. Even if the echo is detected, masking signals may still degrade the accuracy of the sensory estimations, that is the distance and angle, see below. Masking sounds arriving outside the reception window do not interfere with detection.

The **filter-bank receiver** is based on the bat hearing model described in Weißenbacher and Wiegrebe (2003) (*Weissenbacher and Wiegrebe, 2003*). The detector consists of an 80-channel gammatone filter-bank with frequency bandwidths simulating the tuning curve of the inner-ear (*Boonman and Ostwald, 2007*; *Wittekindt et al., 2005*). The impulse response of each channel in this model is given by Equation 6; where $n$ is the filter order (set to 4), $b$ is the time constant of the impulse-response (set to 0.15 $f_c$), $f_c$ is the center frequency of the channel (Equation 7). The signal is filtered by a lowpass filter, which keeps only the envelope of the signal ('envelope detector'). Note, that some biological models (e.g. *Sanderson et al., 2003*) assume that part of the phase information is available to the bat. We chose to implement a simpler biological model in order to set the lower bound for the problem, as opposed to the cross-correlation model that we tested and provides an upper-bound for the effect of JAR. However, to validate that the alternative biological model does

not change the results, we also implanted the half-wave rectifier with a 10kHz LPF model and confirmed that it did not influence our results in one context (see *Figure 2—figure supplement 6*).

$$g(t) = at^{n-1}e^{-2\pi bt}cos(2\pi f_c t) \tag{6}$$

$$f_{ck} = 5702 \cdot 2^{k/20.9} \tag{7}$$

Then, we shift each channel in time to compensate for the delay time between the emission time and the response time of each channel, according to the chirp's slope. Finally, we sum the time-compensated filter-channels and look for peaks in the integrated signal. Prominent peaks that are higher than the detection-threshold, set to 7 dB-SPL, are referred to as potentially detected echoes, and their distance from the bat is estimated relative to the peak detection-time, see *Figure 2—figure supplement 4* panel C. Note that these peaks might be a result of desired echoes or masking signals and their amplitude will be determined simply by running them through the filter-bank model.

Like with the correlation-receiver, for each transmitted call, we implemented the filter-bank receiver twice: (a) only on the echoes from the bat's own emitted call, and (b) on both masking signals and echoes. We compared the peaks detected in (a) with the peaks in (b) and then defined the following criteria: **a jammed signal** is a peak that was detected in (a) but not in (b). The **time-estimation error** is the difference between the estimated peak-time in (b) and the actual received time of the reflected echo. If the first peak in (b) was not detected in (a) and was the first detected peak (in both a and b), the bat mistakenly decided to pursue a 'fake target' (i.e., a masking signal). This case was defined as a **false-alarm**. Note that 'correlation-receiver' assumes that the bat can differ between desired prey echoes and masking signals and echoes from conspecifics (*Amichai et al., 2015*; *Yovel et al., 2009*). On the other hand, the 'filter-bank receiver' did not assume that the bat can discriminate between desired echoes and masking signals and the simulated bats thus pursued the first detected echo (which is, in this case, a masking signal). Therefore, false-alarms were only applicable to the filter-bank receiver.

The SNR (Signal to Noise and interference Ratio) is calculated for each detector by *Equation 8*.

$$SNR = \begin{cases} \frac{max(SelfCorr)}{max(CrossCorr) + NoideLevel}, & for\,Correlation\,Detector \\ \frac{max(Pd_r)}{max(pd_{mask})_{t \in detectiontime} + NoiseLevel} & for\,Filter-Bank\,Detector \end{cases} \tag{8}$$

To analyze the effect of call duration on performance, we modified the model by implementing the correlation-detector in the detection process too, before the masking calculations. Here, the received echo was first correlated with the emitted call, and then it was compared to a detection-threshold. The detection threshold for the correlation was set to 15 dB-SPL, which equals to the maximum of the autocorrelation of n 8 ms 'search' call.

After the detection process, the bat estimates the range and the Direction Of Arrival (DOA) of the reflecting object, based on all of the received signals (echoes and masking signals). The range estimation is based on the acoustic two-way time-travel of the signal with an error (*Equation 9*), comprised of two elements: the bat's accuracy in measuring time, and a noise term which reduces with increasing SNR (Equation 8, calculated using either the Correlation or the Filter-bank model). For simplicity, because all bat calls are FM-chirps, we use an error that is independent of the signal's parameters: the term $\frac{k_1}{SNR}$, where $k_1$ is a coefficient set to scale the error to values of ±1cm at SNR of 10dB, based on behavioral studies (*Denny, 2004*; *Popper and Fay, 1995*; *Masters and Raver, 1996*; *Moss and Schnitzler, 1989*). The independence of the signal's parameters is reasonable because all bat signals (calls, echoes, and masking signals) in our simulation were similar. The term $Time_{Res}$ (in *Equation 9*) is sampled from a Gaussian distribution with a mean ± SD of 0±50 microseconds, equivalent to a range error of 0.85 cm, which is a low boundary estimation of bats capabilities to measure distance (*Genzel et al., 2019*). 'c' is the speed of sound: 343m/s.

$$Range_{error} = \sqrt{\left(\frac{k_1}{SNR}\right)^2 + (0.5 \cdot c \cdot Time_{Res})^2} \tag{9}$$

The estimation error of the DOA, see *Equation 10*, includes an error which depends on the DOA (*i.e.* $k_3 + k_4 \cdot sin(\phi)$ ), set such that the error equals 1.5° at 0° DOA, and error of 10° at 90° DOA, and

a Gaussian noise term: 0°±1° (mean ± SD) at SNR=10dB, see $\frac{k_2}{SINR}$ in ('Hearing by Bats', Chapter 3.1, **Popper and Fay, 1995**; **Simmons et al., 1983**). **Figure 2—figure supplement 5** depicts the histogram of the consequential range errors and DOA errors. Note, in our model, we did not simulate the HRTF (Head Related Transfer Function).

$$DOA_{error} = \sqrt{\left(\frac{k_2}{SNR}\right)^2 + (k_3 + k_4 \cdot sin(\phi))^2} \tag{10}$$

In general, our model intentionally underestimates the bats' actual performance, and thus real bats are likely to cope with acoustic masking even better than our simulated bats: (1) we simulated monaural bats, while real bats use two ears with spatial selectivity (**Griffin et al., 1963**). We also did not model the effect of reducing masking achieved thanks to directional hearing and referred to as spatial unmasking. Spatial unmasking can improve the detection in the presence of off-axis maskers by up to ca. 30 dB in the Big Brown Bat, *Eptesicus fuscus* (**Sümer et al., 2009**). (2) We assumed a low detection threshold (0 dB-SPL), thus the bats were more susceptible to masking (see **Figure 4** C2). (3) We chose long backward and forward masking windows (3 ms, 1 ms), and low jamming thresholds: 0 dB for backward masking, 5 dB for forward masking (e.g. a masking signal that was received first, even if it is 5 dB lower than the desired echo, will completely jam it). (4) The model implied a pulse-by-pulse detection and estimation strategy with no memory, therefore, temporal miss-detections caused by momentary jamming had a very substantial effect on hunting attempts.

## Acoustics calculations

The estimated intensities of the reflected echoes based on the sonar/radar equation (**Mazar, 2016**, pp. 196–198), shown in **Equation 11**, angles and distances are defined according to **Figure 1—figure supplement 3**.

$$P_r = P_t \cdot \frac{G_t(\phi_{target}, f) \cdot G_r(\phi_{target}, f)\lambda^2}{(4\pi)^3 D^4} \cdot 10^{-2\alpha att(f)/10 \cdot (D-0.1)} \cdot \sigma(f) \tag{11}$$

Where:

$P_r$: Source level of the received signal [SPL]
$P_t$: Received level of the transmitted call [SPL]
$\phi_{target}$: Angle between the bat direction and the prey [rad]
$G_t(\phi, f)$: Gain of the transmitter (mouth of the bat), as a function of azimuth and frequency (*f*)
$G_r(\phi, f)$: Gain of the receiver (ears of the bat)
D: Distance between the bat and the object [m]
$\alpha_{att}(f)$: Atmospheric absorption coefficient for sound [dB/m]
$\sigma(f)$: Sonar cross-section of the target [m$^2$]
$\lambda$: Wavelength of the signal [m]

The source level ($P_t$) of a search call equals 110 dB-SPL at a distance of 0.1 meters from the source (**Simmons et al., 1983**). $P_t$ during the hunt varies according to the distance to the prey (see **Table 1**). The transmission gain of the bat's mouth Gt($\phi$, f) is modeled by a circular piston source (**Strother and Mogus, 1970**; **Kounitsky et al., 2015**; **Jakobsen and Surlykke, 2010**), given in Equation 12. The directivity of the emitted call depends on the ratio between the wavelength of the signal ($\lambda$) and the radius of the emitter (mouth), 'a' (set to 3 mm [**Kounitsky et al., 2015**]); J$_1$ is the Bessel function of the first order and k= 2$\pi$/$\lambda$. G$_0$ of the mouth (at the head direction) is set to 1, matching the measurements of intensities of bat's calls from a distance of 0.1m.

$$G(\lambda, \phi) = G_0 \cdot [\frac{2J_1(K \cdot a \cdot sin\phi_{target})}{K \cdot a \cdot sin\phi_{target}}]^2 \tag{12}$$

$$G_0 = \begin{cases} 1 & : mouth\,gain \\ \frac{4\pi}{\lambda^2}\pi a^2 & : ear\,gain \end{cases} \tag{13}$$

To estimate the gain of the ears (G$_r(\phi)$ in **Equation 11**), we modeled them as circular planes,

using the piston model (*Kuc, 1994*), with 'a' (the ear radius) set to 7 mm, matching *Pipistrellus kuhlii*'s ear size (*Keeley et al., 2018*). $G_0$, the maximum gain of the ear, is defined by *Equation 13* (*Mazar, 2016 equation 5*.36, pp. 181), where A is the geometric area of the ear. Since *Pipistrellus* bats do not move their ears, this is a reasonable estimate.

The original piston model is symmetric: the back-lobe is equal to the front-lobe. In our model, bats receive signals from the back hemisphere too, therefore we modified the piston model for the back hemisphere (for transmission and reception), and it is shaped by the piston model with additional attenuation of 0–20 dB, increasing linearly (in dB) from ±90° to ±180°, relative to the bat. The modified piston model for ear and mouth are illustrated in *Figure 1—figure supplement 4*.

$\alpha_{att}$ in the sonar equation (*Equation 11*) is the atmospheric absorption coefficient, set to temperature 20℃ and humidity 50%, is approximated by *Equation 14* (*Kuc, 1994*).

$$\alpha_{att} = 3.8 \cdot 10^{-2} f_{[kHz]} - 0.3 \quad \left[\frac{dB}{m}\right] \tag{14}$$

$\sigma$, the Radar (sonar) Cross-Section (RCS, or 'target strength') of the moth, is modeled as a disc with a radius ('r') equals 2 cm, equally reflecting in all directions. We apply the approximation of the RCS for this type of reflector (*Pouliguen et al., 2008*), given in *Equation 15*, where A is the geometric area of the disc ($A=\pi r^2$). 'r' was set to 2cm, simulating a moth's wing-length. This approximation is in line with measurements of the target strength of medium-sized insects (*Boonman et al., 2013*; *Figure 1*; *Kober and Schnitzler, 1990*).

$$\sigma = \frac{4\pi A^2}{\lambda^2} \tag{15}$$

The receiving time-delays of the echoes are the 2-way duration of the signals propagating from the emitter to the target, see *Equation 16*, where, 'c' denotes sound velocity, D is the distance between bat and target, and t is the overall travel time from the emission to the reception.

$$t = 2D/c \tag{16}$$

The received level of masking signals ($P_{mask}$), i.e. echolocation calls transmitted by conspecific bats ($t_x$) and received by the bat ($r_x$), is calculated by *Equation 17*, where $P_t$ is the source level of the masking call, and the angles are depicted in *Figure 1—figure supplement 3*.

$$\begin{aligned} P_{mask} \quad &= P_t \cdot \frac{G_t(\phi_{t_x r_x})}{4\pi D^2} \cdot \frac{G_r(\phi_{r_x t_x})}{4\pi} 10^{\alpha att \cdot (D-0.1)} \\ &= P_t G_t(\phi_{t_x r_x}) G_r(\phi_{r_x t_x}) \cdot \left(\frac{\lambda}{4\pi D}\right)^2 10^{-\alpha att \cdot (D-0.1)} \end{aligned} \tag{17}$$

In addition to its own echoes (*Equation 11*) and conspecific calls (*Equation 17*), each bat also receives echoes that are reflections off objects from conspecific calls. The intensities of these potentially masking sounds are estimated by modifying *Equation 11* (sonar equation), with fitting angles and distances. This calculation is described in *Equation 18*, and the angles are defined in *Figure 1—figure supplement 3*, arrows 3–4.

$$P_{echoes\,from\,Masking} = P_t \cdot \frac{G_t(\phi_{tx} \cdot G_r(\phi_{rx})\lambda^2}{(4\pi)^3 D_{tx}^2 D_{rx}^2} 10^{-\alpha att \cdot (D_{tx}+D_{rx}-0.2)} \cdot \sigma(f) \tag{18}$$

In nature, bats must also deal with the background or clutter echoes reflected off nearby objects, such as vegetation or cave-walls. To examine how background clutter echoes influence hunting in a group and with and without using a JAR, we ran another set of simulations where we also simulated the echoes of the borders of the arena. The borders were modeled as an array of points separated by 5.7 degrees, relative to the bat's emission. We calculated the timings and intensities of each echo reflected off the points, using the sonar equation (*Equation 11*), with a target radius of 3 cm (similar to the size of a leaf). We gradually increased the reflectors' target strength by 0 to 40 dB at the frequency of 40 kHz, simulating a variety of shapes and materials of reflectors. We then summed (using a non-coherent summing) the echoes of all points within an angle of ±90 degrees relative to the bat's emission. We defined a clutter masking event whenever the summed clutter echo is received during the reception window time (see above and *Figure 2—figure supplement 3*) with higher level

than the echo from the prey item. In a clutter masking event, the bat does not detect the prey item at all. The results are depicted in *Figure 3—figure supplement 2*. All other trials were performed without any background clutter.

## Data analysis and statistics

In each scenario that we simulated, we tested the effect of the varied parameters on hunting performance using ANCOVA, ANOVA, or multiple regression, depending on the type of the parameters (e.g. continuous or categorical) and their number. The Statistics were calculated using Standard Least Squares method with JMP 14 and MATLAB R2018b.

For testing the significance of the masking effect, we used a different procedure. Because the masking effect is a ratio (see *Equation 1*) we had to compute its SD. Thus, for each set of conditions, we estimated the standard deviation of the masking effect, using *Equation 19* (*Stuart and Ord, 1998*). To determine whether the masking effect significantly dependent on the tested parameters, we simulated 100 points in each scenario with the average and SD calculated above and executed ANCOVA to estimate the F test, for the simulated scenario. We repeated this process 1000 times in each scenario and used the average results of the F tests and p-values as the statistics. This process of repetitions was executed only for the statistics of the masking effect.

$$
\begin{aligned}
(1)\ & E\left(\frac{u}{v}\right) = \frac{E_u}{E_v} \\
(2)\ & STD\left(\frac{u}{v}\right) = \frac{E_u}{E_v} \cdot \sqrt{\left(\frac{\sigma_u^2}{E_u^2} + \frac{\sigma_v^2}{E_v^2}\right)}
\end{aligned}
\tag{19}
$$

We also evaluated the 'Jamming Probability' as the proportion of pursued prey echoes (echoes of prey the bat chose to pursue) that are blocked by masking signals (*Equation 20*). Note, that this ratio is not the proportion of all the jammed echoes to all detected echoes, because for each call there can be several detected echoes from different prey items, but only one (maximum) is pursued.

$$
Jamming\ Probability \equiv \frac{total\ pursuit\ jamming\ echoes}{total\ number\ of\ echoes\ from\ hunted\ prey}
\tag{20}
$$

## Source data

Source data and code summary table: The attached Source data and code summary table summarizes the source-data and source code used to produce the figures of this research. The source-data and source-code are attached as supplementary data, hereafter. **readme Bat Simulation**: The file '**readme Bat Simulation**' explains how to execute the attached GUI (Graphical User Interface) and MATALB code which generate the sensorimotor predator-prey simulation.

## Additional information

### Funding

| Funder | Grant reference number | Author |
| --- | --- | --- |
| Office of Naval Research Global | N62909-16-1-2133-P00003 | Omer Mazar |

The funders had no role in study design, data collection, and interpretation, or the decision to submit the work for publication.

### Author contributions

Omer Mazar, Software, Formal analysis, Funding acquisition, Validation, Visualization, Methodology, Writing - original draft, Writing - review and editing; Yossi Yovel, Conceptualization, Resources, Supervision, Funding acquisition, Validation, Investigation, Methodology, Project administration, Writing - review and editing

### Author ORCIDs

Omer Mazar (iD) https://orcid.org/0000-0001-9763-4621

Decision letter and Author response
Decision letter https://doi.org/10.7554/eLife.55539.sa1
Author response https://doi.org/10.7554/eLife.55539.sa2

## Additional files

### Supplementary files

• Source code 1. The source code and MATLAB function requirt to run the simulation. See readme Bat Simulation files for more details.

• Source data 1. 'readme Bat Simulation'explains how to execute the attached GUI (Graphical User Interface) and MATALB code which generate the sensorimotor predator-prey simulation.

• Source data 2. Summary table of all the figures and the relevant data-tables.

• Transparent reporting form

### Data availability

All data generated during this study are included in the manuscript and supporting files. Source code files are uploaded with a Graphical User Interface and a readme file for explanation.

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
