## [Decision Letter]

Thank you for submitting your article "A sensorimotor model shows why a spectral jamming avoidance response does not help bats deal with jamming" for consideration by *eLife*. Your article has now been reviewed by three peer reviewers, and the evaluation has been overseen Andrew King as the Senior Editor and Reviewing Editor. The following individuals involved in review of your submission have agreed to reveal their identity: Dieter Vanderelst (Reviewer #1); James A. Simmons (Reviewer #3).

The reviewers have discussed the reviews with one another and the Reviewing Editor has drafted this decision to help you prepare a revised submission.

Summary:

This paper addresses an important and long-standing question in the field of echolocation, namely whether bats require a jamming avoidance response to avoid mutual interference among multiple bats flying in the same location while hunting for flying insects. The authors' study provides an excellent example of the application of computational modeling, supported by parameterization from the large body of experimental evidence in bat echolocation, to address a complicated question about sensory ecology. It is well established under laboratory conditions that bats change their echolocation frequencies in the presence of other bats or when a tonal jamming signal is delivered at frequencies at or near the ending frequency of their biosonar calls. The principal conclusion from this study is that these frequency shifts do not materially improve prey capture, raising the possibility that the jamming avoidance response is a solution proposed to solve a largely non-existent problem.

Revisions:

Although all three reviewers were generally enthusiastic about this paper, they raised several concerns and made a number of suggestions that will need to be addressed before it can be considered suitable for publication. Most of these points are relatively minor, and concern issues of clarity and presentation. However, it was pointed out that the manuscript does not adequately cite previous literature and that the modelling results are over-interpreted because of the availability, at least in natural environments, of other cues that help to discriminate between other bat sounds and echoes. In addition, some questions were raised about the model.

1) The manuscript includes unsupported claims of novelty (start of Discussion) and does not cite all the relevant literature. Several papers have previously quantified the probability of jamming (Jarvis et al., 2013; Lin and Abaid, 2015; Beleyur and Goerlitz, 2019,). Moreover, a paper (Cvikel et al., 2015) from the same lab has investigated the question of decreasing hunting performance, albeit with a simpler computational model lacking the sensory detail in this current work. One of the primary conclusions “..that jamming is less of a problem than previously suggested” has also previously been shown by Beleyur and Goerlitz, 2019, with a similar approach of biologically parameterized detailed sensory simulations. This work should be cited at appropriate places in the manuscript, as this will help readers to contextualize the findings.

2) The studies that demonstrate putative frequency jamming avoidance responses (JARs) in bats did not go further with interpretation than saying that JARs are just one of several dimensions along which bats could distinguish echoes of their own broadcasts to discriminate against similar but not identical sounds of other bats. For example, changing pulse times and FM sweep slopes provide ample opportunities for discriminating among multiple other bat sounds and echoes. The modeling work in this manuscript establishes that the frequency JAR is not by itself a critical dimension for avoiding mutual interference. For this reason, the work is an eminently valuable contribution, but the interpretation is too strong since other cues are typically available too. Furthermore, in at least one study, the shifts in frequency are related to pulse-echo ambiguity or self-jamming in complex scenes (Hiryu et al., 2010). This is a much more difficult problem than mutual interference in groups of bats, so the authors should be guarded in their interpretation of the contribution to JARs to the ecology of echolocating bats and specifically why they occur if they do not materially enhance prey interception.

3) In the manuscript's filterbank model, the envelope detector is described as removing phase information and is similar to several loosely auditory-inspired models of echolocation (Wiegrebe, 2008; Peremans and Hallam, 1998; Boonman and Ostwald, 2007). However, a more biologically realistic model with half-wave rectification and 10 kHz low-pass filtering performs the same as a crosscorrelating receiver (Sanderson et al., 2003). In general, there is a lot of work on time-frequency methods that are equivalent to matched filtering, but the authors choose a model that removes phase information even though behavioral tests of both bats and dolphins suggest that echo phase may be perceived. It would be helpful if the authors give their reason for this limitation of their filterbank model, particularly as inclusion of phase information would likely strengthen the study's findings.

4) When the authors model localization errors due to jamming, they do this by assuming that a lower SNR leads to larger errors (Equations 9 and 10). This is appropriate, but it is not the whole picture. Interference between echoes could obscure spectral cues needed for localization. This is not taken into account in this model. The authors should at least acknowledge the possibility of spectral interference and state why they do not model it explicitly.

5) More attention needs to be paid to the writing as various terms are used loosely, introduced without definition, or used interchangeably.

---

## [Author Response]

Revisions:Although all three reviewers were generally enthusiastic about this paper, they raised several concerns and made a number of suggestions that will need to be addressed before it can be considered suitable for publication. Most of these points are relatively minor, and concern issues of clarity and presentation. However, it was pointed out that the manuscript does not adequately cite previous literature and that the modelling results are over-interpreted because of the availability, at least in natural environments, of other cues that help to discriminate between other bat sounds and echoes. In addition, some questions were raised about the model.1) The manuscript includes unsupported claims of novelty (start of Discussion) and does not cite all the relevant literature. Several papers have previously quantified the probability of jamming (Jarvis et al., 2013; Lin and Abaid, 2015; Beleyur and Goerlitz, 2019). Moreover, a paper (Cvikel et al., 2015) from the same lab has investigated the question of decreasing hunting performance, albeit with a simpler computational model lacking the sensory detail in this current work. One of the primary conclusions “..that jamming is less of a problem than previously suggested” has also previously been shown by Beleyur and Goerlitz, 2019, with a similar approach of biologically parameterized detailed sensory simulations. This work should be cited at appropriate places in the manuscript, as this will help readers to contextualize the findings.

In the resubmitted manuscript we added the references that were missed in the previous version, according to the reviewers’ comments. We discarded the claims for novelty and cited the suggested references at the appropriate places.

2) The studies that demonstrate putative frequency jamming avoidance responses (JARs) in bats did not go further with interpretation than saying that JARs are just one of several dimensions along which bats could distinguish echoes of their own broadcasts to discriminate against similar but not identical sounds of other bats. For example, changing pulse times and FM sweep slopes provide ample opportunities for discriminating among multiple other bat sounds and echoes. The modeling work in this manuscript establishes that the frequency JAR is not by itself a critical dimension for avoiding mutual interference. For this reason, the work is an eminently valuable contribution, but the interpretation is too strong since other cues are typically available too. Furthermore, in at least one study, the shifts in frequency are related to pulse-echo ambiguity or self-jamming in complex scenes (Hiryu et al., 2010). This is a much more difficult problem than mutual interference in groups of bats, so the authors should be guarded in their interpretation of the contribution to JARs to the ecology of echolocating bats and specifically why they occur if they do not materially enhance prey interception.

The argument that there are many possible types of information allowing to discriminate own calls is, actually, the main claim of our manuscript: “bats constantly change their signals according to behavioral phase and distance to nearby objects … their signals are different due to their different behavioral phase”. Our model shows that those differences in signal design are usually enough to discriminate between echoes and conspecifics’ sounds even without applying a spectral JAR. We agree with the reviewer that there is ample information for discrimination, but most previous research focused on spectral JAR and thus this is what we focus on too.

Our model intentionally underestimates the bats’ actual performance, and thus real bats are likely to cope with acoustic interference even better than our simulated bats. For example, the simulated bats sense their environment using one sense only, echolocation. In natural environments, any other sensorial cue helping the bats to discriminate between peers’ sounds and desired echoes will further reduce the impact of jamming on hunting. This also supports our findings that much of the interference that bats suffer from when foraging in a group results from competition over prey and from the need to avoid conspecifics, and not from acoustic masking. In the revised version, we tried to tune our interpretations and removed claims that might be too strong.

Several previous studies reported that bats shift their emission frequencies in response to nearby conspecifics. Researchers have interpreted this behavior as a jamming avoidance response. Importantly, the great majority of these studies found an increase in frequency in the presence of conspecifics, independently of the frequencies of the two nearby bats, or the playback sounds. In this revised paper, we suggest several alternative hypotheses that could explain these findings. We also mention alternative explanations such as avoiding ambiguity, hopefully now conveying better the complexity of the story. Once again, we agree with the reviewer that there are many possible explanations for frequency shifts, but here, we focus on the commonly suggested spectral JAR hypothesis, because it has become central in the discussion although the evidence for it is not robust.

3) In the manuscript's filterbank model, the envelope detector is described as removing phase information and is similar to several loosely auditory-inspired models of echolocation (Wiegrebe, 2008; Peremans and Hallam, 1998; Boonman and Ostwald, 2007). However, a more biologically realistic model with half-wave rectification and 10 kHz low-pass filtering performs the same as a crosscorrelating receiver (Sanderson et al., 2003). In general, there is a lot of work on time-frequency methods that are equivalent to matched filtering, but the authors choose a model that removes phase information even though behavioral tests of both bats and dolphins suggest that echo phase may be perceived. It would be helpful if the authors give their reason for this limitation of their filterbank model, particularly as inclusion of phase information would likely strengthen the study's findings.

We agree that some modeling and behavioral data suggest that the filter-bank does not remove all phase information. In the original study, we chose to use a model that does remove all phase information in order to set a lower bound. The cross-correlation model that we tested provides an upper-bound for the effect of JAR and thus by choosing the simpler biological model we could set the lower bound for the problem. Since both models did not show any advantage to the JAR response, we can safely assume that the more complex biological method would also show the same result. In the revised manuscript, we verified this, by implementing the half-wave rectifier with10kHz LPF model and confirming that it did not influence our results (Figure 2—figure supplement 6).

4) When the authors model localization errors due to jamming, they do this by assuming that a lower SNR leads to larger errors (Equations 9 and 10). This is appropriate, but it is not the whole picture. Interference between echoes could obscure spectral cues needed for localization. This is not taken into account in this model. The authors should at least acknowledge the possibility of spectral interference and state why they do not model it explicitly.

First, regarding the range error, we would like to point that the ranging errors of the filter-bank model are derived directly from the peak detection, without any assumption, therefore in this model, any spectral interference that could affect the results is considered in the model.

Next, regarding the range error when using the correlation receiver, we argue that our model takes into account the spectral notch-related phenomenon mention by the reviewer. Equations 9 and 10 in the manuscript are based on the well-known Cramer Rao Lower Boundary (CRLB): var(τ)≥1ϵN02F2. Where τ is the delay (sec), ϵN02 is SNR and F^2^ is a function of the waveform of the transmitted signal (references ^1,2^). For simplicity, we calibrated the errors according to real experiments performed with real bats (28), both for the ranging errors (of ±1cm at SNR of 10dB) and directional estimations (±1 degree at SNR 10dB, in front of the bat).

To confirm this, we ran several simulations showing that a superposition of two signals (i.e. a desired echo and a masking signal) does not shift the timing of the cross-correlation peak regardless if the resulting spectral notches, see Author response image 1. Supporting this conclusion, Warnecke and Simmons (2016)^3^ indicated that when two signals interfere, the first echo (“glint”) is perceived by bats using the time delay; and not by the spectral pattern of the composite echo.

As for the angular estimation, we agree that spectral interference may impact the estimation of the direction of the signals, as it is derived from the HRTF (Head-related transfer function). In our model, we did not simulate the HRTF. We now mention this directly in the revised manuscript. Nevertheless, the probability of spectral interference between the masking signal and the desired echo is low, because it is a function of the ratio between the signals’ level (referred to as SNR in our work). The maximum spectral interference occurs when the received levels of the two signals are equal, and it decays to a ca. 2dB (peak-to-peak) when the SNR is 10dB (see Author response image 1, compare the top and middle panels). Therefore, spectral interference may increase the errors only when the SNR is between 0dB to ca. 10dB and the probability of it is low (about 15% of the received echoes from the target, with 10 bats and 10 prey items). Moreover, localization errors are significant to hunting mainly during the pursuit, when the bat is getting closer to the prey and the SNR is relatively high. In these cases, the probability of spectral interference reduces. Consequently, even if we underestimate the impact of spectral interference in specific situations, it probably has a minor effect on the final results.

**Author response image 1. sa2fig1:** The Spectral Interference between a desired echo and masking signals with various latencies and SNRs. Left panels depict the cross-correlation between the transmitted call (a linear FM chirp call with a duration of 7milliseconds, 40-70kHz) and the received signal (the desired echo plus the masker). Reference masker signal (black line) in all panels is a Band Pass Filtered noise with the same frequency range as the call (i.e. 40-70kHz). Top panel: One masker with delay 0.3 milliseconds and SNR 0dB. Medium panel: One masker with a delay of 0.3 milliseconds and an SNR of 20dB. Bottom panel: Two maskers with delays of -0.1 (before the desired echo) and 0.4 milliseconds and an SNR of 0dB. Note, the various spectral interferences do not shift the correlation peak of the desired echo (always at time 0) and they do not change the shape of the cross-correlation function.

5) More attention needs to be paid to the writing as various terms are used loosely, introduced without definition, or used interchangeably.

We accepted the comment and defined the terms we are using in the manuscript rigorously.

**References:**

1. Yun, S., Kim, S., Koh, J. and Kang, J. Analysis of Cramer-Rao Lower Bound for Time Delay Estimation using UWB Pulses. 2–6 (2012).

2. Simmons, J. a et al. Delay accuracy in bat sonar is related to the reciprocal of normalized echo bandwidth, or Q. *Proc. Natl. Acad. Sci. U. S. A.***101**, 3638–43 (2004).

3. Warnecke, M. and Simmons, J. A. Target shape perception and clutter rejection use the same mechanism in bat sonar. *J. Comp. Physiol. A***202**, 371–379 (2016).